# Assessing Post-Vaccination Seroprevalence and Enhancing Strategies for Lumpy Skin Disease Vaccination in Korean Cattle

**DOI:** 10.3390/ani14223236

**Published:** 2024-11-12

**Authors:** Geun-Ho Kim, Dae-Sung Yoo, Keum-Suk Chu, Eun-Hyo Cho, Seung-Il Wi, Kyung-Ok Song, Do Kyung Ra, Woo H. Kim, Choi-Kyu Park, Dongseob Tark, Yeonsu Oh, Ho-Seong Cho

**Affiliations:** 1College of Veterinary Medicine & Bio-Safety Research Institute, Jeonbuk National University, Iksan 54596, Republic of Korea; kkh2701@naver.com; 2Jeollabuk-do Institute of Livestock & Veterinary Research, Jangsu 55632, Republic of Korea; chuks1103@korea.kr; 3College of Veterinary Medicine, Chonnam National University, Gwangju 61186, Republic of Korea; shanuar@chonnam.ac.kr; 4Jeollanamdo Veterinary Service Laboratory, Gangjin 59213, Republic of Korea; ddaeng3@korea.kr (E.-H.C.); wsi3866@korea.kr (S.-I.W.); 5Animal Health Division, Jeju 63344, Republic of Korea; songko76@korea.kr; 6Incheon Metropolitan City Institute of Health & Environment, Incheon 22320, Republic of Korea; rara9292@korea.kr; 7College of Veterinary Medicine & Institute of Animal Medicine, Gyeongsang National University, Jinju 52828, Republic of Korea; woohyun.kim@gnu.ac.kr; 8Animal Disease Intervention Center, College of Veterinary Medicine, Kyungpook National University, Daegu 41566, Republic of Korea; parkck@knu.ac.kr; 9Korea Zoonosis Research Institute, Jeonbuk National University, Iksan 54531, Republic of Korea; tarkds@jbnu.ac.kr; 10College of Veterinary Medicine & Institute of Veterinary Science, Kangwon National University, Chuncheon 24341, Republic of Korea

**Keywords:** biosecurity, cattle, lumpy skin disease, lumpy skin disease vaccine, seropositivity, Korea

## Abstract

Lumpy skin disease (LSD) caused by the LSD virus (LSDV) severely impacts the economy of a country and its cattle industry. After the first outbreak in October 2023, South Korea decided to vaccinate cattle nationwide. This study specifically investigated the efficacy of the vaccination campaign against lumpy skin disease (LSD) and analyzed the seropositivity rates of LSD antibodies 2–3 months after vaccination to provide guidance regarding future vaccination strategies and biosecurity measures. In total, 3910 cattle were tested from four Korean provinces, with an overall seropositivity rate of 30.59 percent. Dairy cattle had higher seropositivity rates (42.97%) than Korean native cattle (29.21%). The availability of appropriate restraining facilities and vaccination methods has an influence on vaccine effectiveness. Although vaccination produces an adequate immune response, upgrading vaccination procedures, such as strengthening restraint facilities and employing professionals, can improve outcomes even more. This study offers useful insights into how to improve LSD control strategies in Korea and other affected countries.

## 1. Introduction

Lumpy skin disease (LSD) is an infectious disease affecting livestock caused by the lumpy skin disease virus (LSDV) that infects cattle and water buffalo, causing symptoms such as skin nodules, swollen lymph nodes, high fever, runny nose, watery eyes, and swollen internal organs [1]. It is classified as a Class 1 livestock disease in Korea and as a manageable disease by the World Organisation for Animal Health (WOAH) due to its significant impact on the cattle industry [1,2]. LSDV, the causative agent of LSD, is a virus belonging to the genus Capripoxivirus in the family Poxviridae with a double-stranded DNA genome measuring 270 nm × 290 nm and approximately 151 Kb in length [3]. It has been reported to have a high genetic similarity of 96% with sheeppox virus (SPV) and goatpox virus (GPV), which are susceptible to sheep and goats of the same genus [4,5]. First identified in Zambia, Africa, in 1929 [6], LSD remained endemic to the continent until the late 20th century. The disease then spread beyond Africa, with outbreaks documented in Egypt, Turkey, and Russia, before reaching the Middle East and Europe [7]. Since 2020, the epidemic has spread to Southwest and Southeast Asia, as well as southeastern China, Taiwan, and the Russian Far East [8,9]. 

In South Korea, the emergence of LSD led to nationwide vaccination efforts. Following the first case of LSD, which occurred on 19 October 2023 at a beef cattle farm in Seosan, Chungnam province, a rapid spread ensued, prompting the vaccination of 4.07 million cattle across 93,944 farms within a 21-day period using live attenuated vaccines such as Lumpyvax (MSD Animal Health, Republic of South Africa), LUMPYVAC (VETAL Animal Health Products S.A., Turkiye), and Lumpy Skin Disease Vaccine for Cattle (Onderstepoort Biological Products, Republic of South Africa) [10,11]. Despite the swift action, various issues were raised during the LSD vaccination process, such as the proficiency of the vaccinator and the efficiency of the calibration process, and the challenges of a vaccination efficacy assessment were underscored by the inability to directly measure the protective effects through IgG concentrations. Despite these challenges, international studies have generally shown positive antibody responses after vaccination, indicating that monitoring antibody levels can be an effective way of assessing vaccine performance [12,13,14,15]. This suggests that a further evaluation of vaccination techniques and processes in Korea is necessary to enhance the overall effectiveness of the LSD vaccine strategy. 

The immune response in cattle following immunization with an LSD attenuated live vaccine is characterized by a coordinated activation of both innate and adaptive immune mechanisms. Upon administration, the vaccine’s weakened LSD virus particles are taken up by antigen-presenting cells, processed, and presented via Major Histocompatibility Complex (MHC) molecules, eliciting a strong T-cell response. CD4+ helper T cells and CD8+ cytotoxic T cells play pivotal roles in coordinating and executing the immune defense by activating B cells, which then proliferate and produce virus-specific antibodies. Moreover, the formation of memory B and T cells ensures long-term immunogenicity and a rapid response upon virus re-exposure, which are integral for effective disease control [13,16,17,18].

This study aims to expand upon the global understanding of LSDV immune dynamics by assessing the antibody responses in cattle following a mass vaccination campaign in South Korea. By comparing antibody levels across different cattle breeds from farms in Incheon, Jeonbuk, Jeonnam, and Jeju province, this research seeks to provide insights that could refine vaccine formulations and administration strategies, enhancing the overall effectiveness of LSD control measures in Korea. Through a detailed examination of the immune response mechanisms triggered by the vaccine, this research not only aims to optimize current vaccination approaches but also to contribute to the broader field of viral immunology and vaccine development against poxviruses affecting livestock. 

## 2. Materials and Methods

### 2.1. Study Design and Farm Characteristics

This study aimed to assess the post-vaccination seroprevalence of LSD in Korean cattle, focusing on the effects of farm size, breed, and vaccination practices. A total of 3910 cattle from 373 cattle farms in four provinces (Incheon, Jeonbuk, Jeonnam, and Jeju province) were included. Cattle were dichotomized into large-scale and small-scale farms based on a threshold of 50 cattle, following the criteria for government veterinary services subsidies across the four municipalities. Farms with more than 50 cattle were classified as large-scale, while those with fewer than 50 cattle were considered small-scale farms. 

The study included both Korean native beef cattle and dairy cattle. Important distinctions were observed between farm sizes and breeds in terms of vaccination practices and infrastructure:Large-scale farms: On these farms, vaccinations were typically performed by the farmers themselves. Large-scale dairy farms were more likely to have restraint facilities, which are essential for ensuring the accurate subcutaneous administration of the LSD vaccine.Small-scale farms: On these farms, vaccinations were generally conducted by veterinarians. Restraint facilities were typically not available on these farms, regardless of the breed of cattle. Both small-scale farms and large-scale Korean native beef cattle farms lacked proper restraint facilities, which may have influenced the efficacy of vaccination due to the difficulty in maintaining consistent and accurate vaccine delivery.

### 2.2. Vaccination and Sampling Procedures

Between October and November 2023, the nationwide vaccination campaign utilized three different live attenuated LSD vaccines (Lumpyvax, LUMPYVAC, and Lumpy Skin Disease Vaccine for Cattle). The first case of LSD was reported on a Korean native (Hanwoo) cattle farm on 19 October 2023, with a total of 107 cases confirmed over the subsequent 33 days, from 19 October to 20 November 2023. Following the initial detection, ring vaccination within a 20 km radius of the infected premises commenced on 21 October 2023, and compulsory nationwide vaccination was implemented on 31 October 2023. By 6 November 2023, at the conclusion of the nationwide LSD vaccination campaign, 100% of cattle farms in South Korea had been vaccinated, as indicated by the Ministry of Agriculture, Food, and Rural Affairs (Figure 1). Consequently, a control group without vaccination could not be included in this study. 

Cattle were vaccinated subcutaneously, and serum samples were collected from each animal 2–3 months post-vaccination to assess the presence of antibodies.

For large-scale farms, 16 cattle were sampled per farm, while on small-scale farms, five cattle were sampled per farm. Blood samples were collected and refrigerated, then transported to the respective animal health laboratories, where serum was separated via centrifugation and stored at −20 °C until analyzed.

### 2.3. ELISA 

The antibody test method was performed using a commercially available ELISA kit, ID Screen^®^ Capripox Double Antigen Multi-species kit (IDVET, Montpellier, France), to detect the presence of antibodies. The test results were judged as negative if the S/P (%) value was less than 30 and positive if the S/P (%) value was more than 30 according to the manufacturer’s judgement criteria by comparing the negative/positive control absorbance values [10,19]. 

### 2.4. Statistical Analysis

Univariable chi-square tests were initially conducted to screen potential predictors of LSD seropositivity using GraphPad Prism 8 (GraphPad Software Inc., La Jolla, CA, USA) to evaluate the statistical significance of the association between antibody test results for the LSD vaccine (classified as positive or negative) and herd size. Variables that showed significance in the univariable analysis were then included in a multivariable Bayesian hierarchical logistic regression model to account for confounders and evaluate the independent associations between farm characteristics and seropositivity as follows:$$LSD\;seropositivity_{ij}\sim B\left(\pi_{ij}\right),\;logit\left(\pi_{ij}\right)=\alpha +\beta_{1}*f\left(farm\;size\right)+\beta_{2}*f\left(farm\;produciton\;type\right)+random\;effect({farm}_{j})\;random\;effect\left(cattle\;{farm}_{j}\right)\sim N\left(\mu_{j},{\sigma^{2}}_{j}\right),j:372\;cattle\;farms$$
where f refers to dichotomized variables large-scale and beef cattle farm, and i refers to cattle within a farm j; in turn, LSD seropositivity_ij_ corresponds to the outcome of LSD antibody test on cattle i raised in farm j.

The model took into account the farm-level variability and used three chains in the Markov Chain Monte Carlo (MCMC) process, with a burn-in of 4000 iterations and a thinning rate of 10. Each posterior distribution used 50,000 iterations. Non-informative distribution was assigned to the prior of the regression coefficients in the model. The Bayesian inference was performed using the R2jags package version 0.8-5 in R software version 4.4.1 [20].

## 3. Results

### 3.1. LSD Vaccine Antibody Positivity Rates in Cattle and Cattle Farms by Municipality

The results of the vaccine antibody positivity of individual and farm-level beef and dairy cattle in the four local governments are shown in Table 1 and Figure 2. Of the 3910 animals tested, 1196 (30.59%) were positive for LSD vaccine antibodies. This included 1028 (29.21%) of 3519 beef cattle and 168 (42.97%) of 391 dairy cattle. The difference in positivity rates between beef and dairy cattle was statistically significant (chi-square df = 31.35, *p* < 0.001). The positivity was consistent across the four municipalities. The farm-level LSD vaccine antibody positivity was determined by defining positive farms as those with at least one seropositive animal. Using this definition, 241 out of 373 farms (64.61%) were positive. Among them, 27 out of 32 dairy cattle farms (84.38%) were positive, compared to 214 out of 341 beef cattle farms (62.76%). This difference in farm-level positivity between dairy and beef cattle farms was statistically significant (chi-square df = 5.980, *p* < 0.05). Similar to the individual animal results, the positive rate for farms by the municipality was consistent, with a higher frequency observed in dairy cows.

### 3.2. Seropositivity of LSD Vaccine Antibodies Across Farm Sizes and Cattle Breeds

In the univariable analysis, there were no statistically significant differences in seropositivity for LSD vaccine antibodies between large-scale and small-scale farms, with levels being similar (23.57 ± 22.57% vs. 23.60 ± 25.55%, *p* > 0.05), as illustrated in Figure 3.

However, dairy cattle showed a significantly higher antibody positivity (29.43 ± 30.61%) than Korean native beef cattle (23.02 ± 23.33%) (*p* < 0.05), as indicated in Figure 4. 

A further exploration of antibody positivity by farm size within each breed revealed that beef cattle had a comparable seropositivity between large-scale (21.97 ± 20.79%) and small-scale farms (24.09 ± 25.69%). In contrast, dairy cattle on large-scale farms exhibited a substantially higher seropositivity (43.30 ± 33.39%) compared to those on small-scale farms (19.21 ± 24.51%) (*p* < 0.01), as shown in Figure 5.

### 3.3. Results of Multivariable Analysis 

However, as univariable analyses do not account for potential confounding factors, it was necessary to confirm these findings through a multivariable analysis, controlling for variables such as farm size and vaccination practices. In the final multivariable model, after adjusting for breed and other confounding variables, farm size was not significantly associated with LSD seropositivity (OR: 0.97, 95% CI: 0.77–1.22), confirming that the impact of farm size on seropositivity was minimal when other factors were taken into account. The multivariable model showed that farms with dairy cattle had significantly higher odds of testing positive for LSD antibodies compared to those with Korean native cattle (OR: 1.82, 95% CI: 1.27–2.60), as summarized in Table 2.

### 3.4. Adjusted Odds Ratio of Seropositivity Following LSD Vaccination in Korean Cattle Farms

After the univariable screening, a multivariable Bayesian hierarchical logistic regression model was constructed to account for confounding factors and provide a more accurate estimate of the association between cattle breed, farm size, and seropositivity. The results, as shown in Table 2, indicate that dairy cattle were significantly associated with higher odds of LSD antibody positivity compared to Korean native cattle (OR: 1.82, 95% CI: 1.27–2.60). This model also confirmed that farm size, when adjusted for breed and vaccination practices, was not a significant predictor of seropositivity, further emphasizing the importance of farm infrastructure in vaccine delivery.

The Gelman–Rubin convergence diagnostics indicated that the model parameters had converged (MCMC values < 1.1), ensuring the robustness of these findings. These results reinforce the earlier observation that large-scale dairy farms, with proper restraint facilities, are more effective in generating higher antibody responses, while both small-scale farms and large-scale Korean native beef farms without such infrastructure show comparatively lower seropositivity.

## 4. Discussion

LSD poses a significant threat to the global cattle industry, resulting in substantial economic losses due to reduced productivity, trade restrictions, and the cost of managing outbreaks. In response to the first outbreak of LSD in Korea in 19 October 2023, a nationwide mass vaccination campaign was conducted. This study aimed to evaluate the post-vaccination antibody seropositivity in cattle across four provinces in Korea and examine the effects of farm size, breed, and vaccination practices on seroconversion. Our findings highlight the critical role that these factors play in determining vaccine efficacy, with the breed emerging as a significant factor, while farm size showed no significant association after controlling for other variables. 

Following LSD vaccination, antibody levels typically rise around 10–15 days post-vaccination (DPV), peak around 30 DPV, and gradually decline thereafter. While the presence of detectable antibodies generally indicates successful vaccination and protection against LSD, it is important to note that some vaccinated cattle may develop protective immunity without detectable antibody production [21,22,23,24].

In Korea, both foot-and-mouth disease and LSD vaccines are administered based on farm size. Large-scale farms (>50 heads) are typically vaccinated by farmers, while small-scale farms (<50 heads) receive veterinary support for vaccination [25]. This approach, while logistically practical, introduces variability in vaccination technique, particularly for the subcutaneous LSD vaccine. Unlike the intramuscular FMD vaccine, subcutaneous administration requires proper restraint and handling of the animal to ensure accurate vaccine delivery.

One of the key findings from the multivariable Bayesian hierarchical logistic regression model was that farms with dairy cattle had significantly higher odds of vaccine-induced antibody positivity compared to farms with Korean native beef cattle (OR: 1.82, 95% CI: 1.27–2.60). However, this difference likely reflects the presence of restraint facilities in large-scale dairy farms, which enable more accurate vaccine administration. Both large-scale farms with Korean native beef cattle and small-scale farms, which often lack these facilities, exhibited lower seropositivity. This indicates that farm infrastructure, rather than breed alone, plays a critical role in vaccine efficacy.

Interestingly, while univariable analyses initially suggested no significant differences in seropositivity between large- and small-scale farms, the multivariable model confirmed that farm size did not have a significant impact on seropositivity after controlling for breed and other factors (OR: 0.97, 95% CI: 0.77–1.22). This finding suggests that large-scale farms do not necessarily achieve better vaccine efficacy unless they are equipped with proper infrastructure, such as restraint facilities, to support accurate vaccine administration. The Gelman-Rubin convergence diagnostics confirmed the robustness of the multivariable model, ensuring that the observed associations are reliable and not due to chance.

The role of vaccination practices and the use of restraint facilities appeared to be a significant contributor to the observed differences in seropositivity between dairy and Korean native cattle. Dairy farms, which are more likely to use restraint facilities, achieved higher seropositivity, suggesting that accurate subcutaneous vaccine administration is more likely to be achieved with proper animal restraint. This observation is consistent with previous research showing that inadequate restraint or handling can compromise vaccine delivery and reduce efficacy [17]. On beef farms, particularly smaller operations, the lack of restraint facilities may have contributed to lower antibody positivity, highlighting the importance of improving vaccination techniques, especially in environments where handling infrastructure is limited. 

Comparative analysis of global LSD vaccination efforts and outcomes.

LSD vaccination efforts vary globally, with each country adapting its approach based on local agricultural practices, economic considerations, and veterinary infrastructure. Notable examples include Serbia, Belgium, and Ethiopia, each employing distinct strategies and achieving varying degrees of success. Our findings are consistent with those international reports of LSD vaccination. Serbia reported seropositivity of 33.77% after initial vaccination, which increased to 57% after a booster dose one month later [12]. This underscores the potential benefits of a booster in enhancing immune response. Belgium reported a seropositivity of approximately 30% at 20 days post-vaccination, highlighting the challenges of achieving high seropositivity rates with a single vaccine dose. Ethiopia recorded 41.65% seropositivity 30 days after vaccination, illustrating a relatively effective vaccination rollout despite logistical challenges [13,14]. The overall seropositivity in our study (30.59%) falls within this range, suggesting that Korea’s vaccination efforts achieved results comparable to international standards, despite the rapid rollout and the use of three different live attenuated vaccines.

These cases show varying success rates and highlight the importance of booster doses, the timing of vaccinations, and the logistical capacities of each country. Like Korea, these countries have adapted their vaccination strategies to meet their specific needs and challenges.

Despite these successes, vaccination alone is not sufficient to control LSD outbreaks, as comprehensive control measures are necessary to prevent further transmission. LSD is primarily spread through blood-feeding insects, such as mosquitoes and ticks, as well as through direct contact with infected animals. Environmental factors, such as the density of vectors and the movement of infected animals, significantly influence the spread of the disease. Therefore, in addition to vaccination, biosecurity measures that limit the movement of animals and control insect vectors are essential for curbing LSD transmission [8,26,27,28].

Moreover, while seropositivity indicates a successful humoral immune response, it is important to recognize that cell-mediated immunity plays a crucial role in protecting cattle against LSD. Although ELISA-based antibody tests are useful for assessing the vaccine response, some animals may develop protective immunity without producing detectable antibodies [29,30]. Future research should focus on evaluating both humoral and cellular immune responses to provide a more comprehensive understanding of vaccine efficacy. This approach would better inform future vaccination strategies and improve the control of LSD in Korea.

## 5. Conclusions

This research sheds light on the dynamics of vaccine-induced seropositivity against LSD in Korea, underlining the varying influences of breed, farm infrastructure, and vaccination techniques. This study affirmed that dairy cattle generally achieve higher seropositivity, which can be attributed to more consistent vaccination practices facilitated by better infrastructure, particularly in terms of restraint facilities. Contrary to the initial assumption, farm size did not significantly impact seropositivity when other factors were accounted for, emphasizing the need for uniform vaccination approaches across different scales of operation.

Additionally, the comparative analysis with international LSD vaccination data highlighted the nuanced strategies and outcomes observed globally, reinforcing the effectiveness of Korea’s rapid response despite the challenges posed by employing multiple vaccine types. Moving forward, improving vaccine delivery and expanding biosecurity measures will be crucial in enhancing the overall efficacy of LSD vaccination programs. These strategies should be particularly targeted at smaller farms that currently face logistical constraints. By focusing on these areas, future interventions can be better tailored to the diverse needs of the cattle industry, ensuring a more robust defense against LSD outbreaks both nationally and internationally.

## Figures and Tables

**Figure 1 animals-14-03236-f001:**
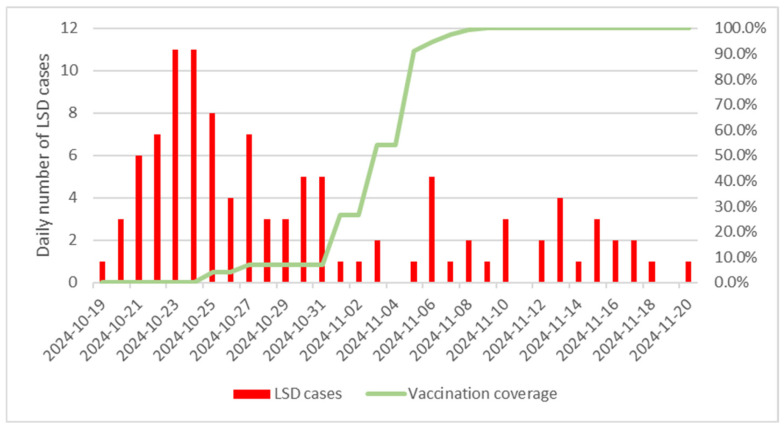
Epidemic curve of the lumpy skin disease (LSD) outbreak and vaccination coverage in South Korea. The green line indicates the cumulative nationwide vaccination coverage among cattle farms, while the red bars represent the daily number of reported LSD cases on cattle farms.

**Figure 2 animals-14-03236-f002:**
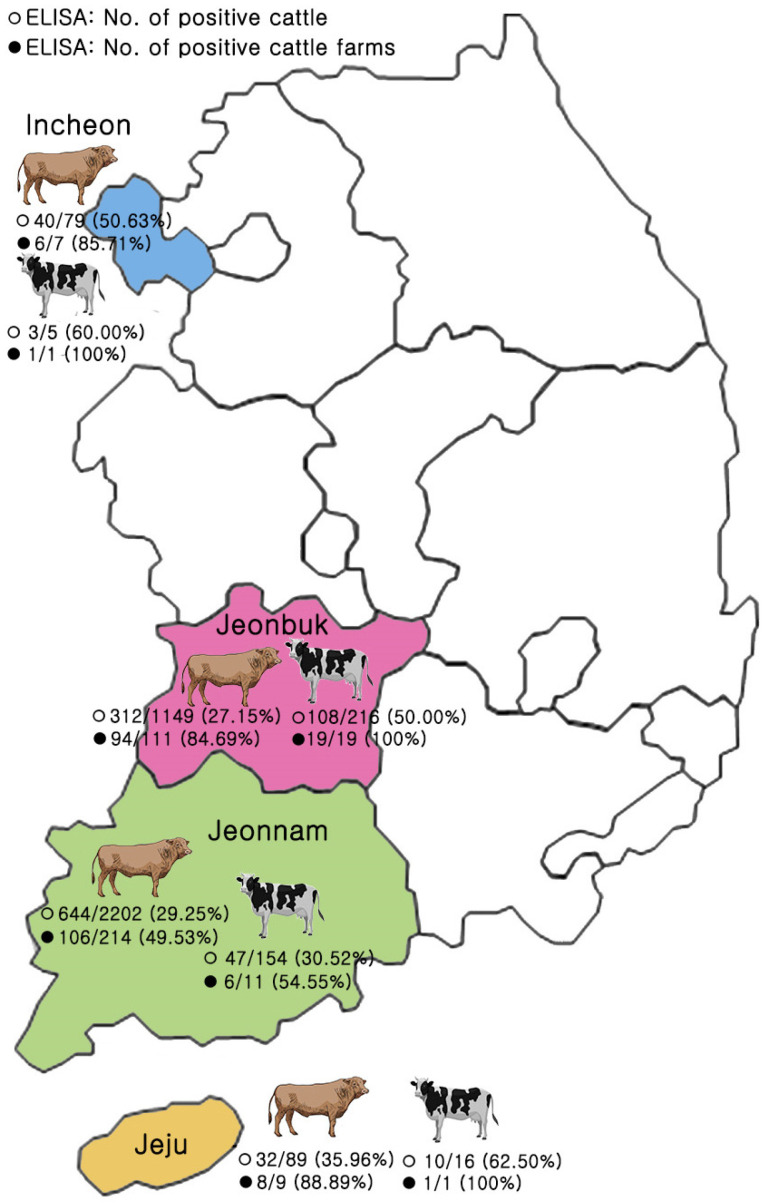
Geographical distribution of seropositivity by ELISA after lumpy skin disease vaccination in Korean native cattle and dairy cattle in Korea.

**Figure 3 animals-14-03236-f003:**
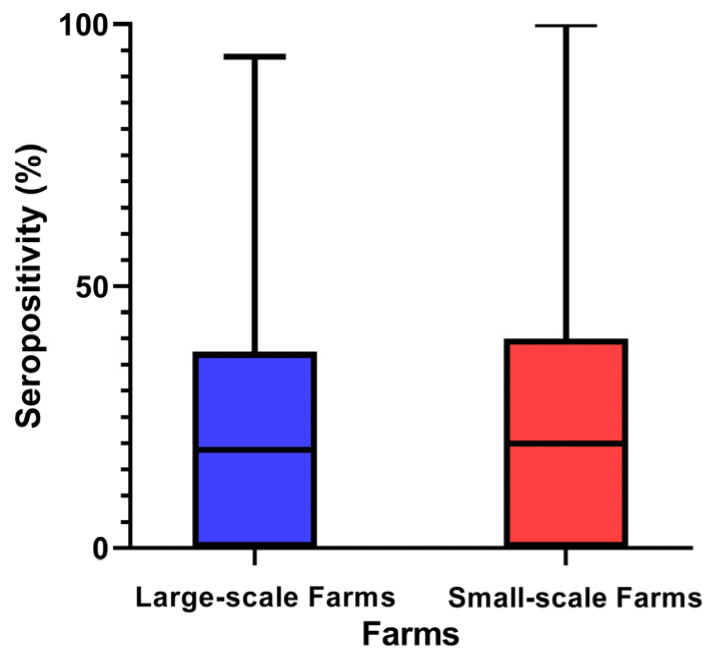
Comparison of seropositivity in Korean cattle farms by farm scale.

**Figure 4 animals-14-03236-f004:**
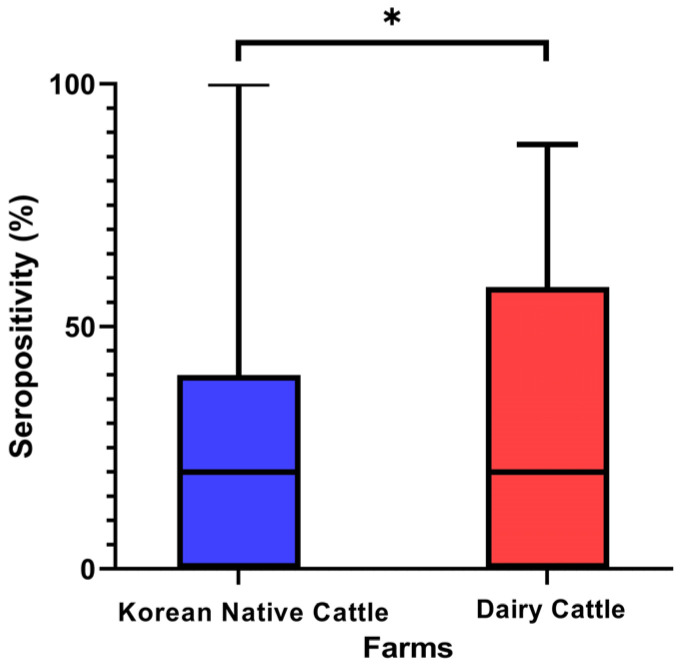
Comparison of seropositivity in Korean cattle farms by breed: Korean native and dairy cattle after LSD vaccination. * Statistically significant differences (*p* < 0.05).

**Figure 5 animals-14-03236-f005:**
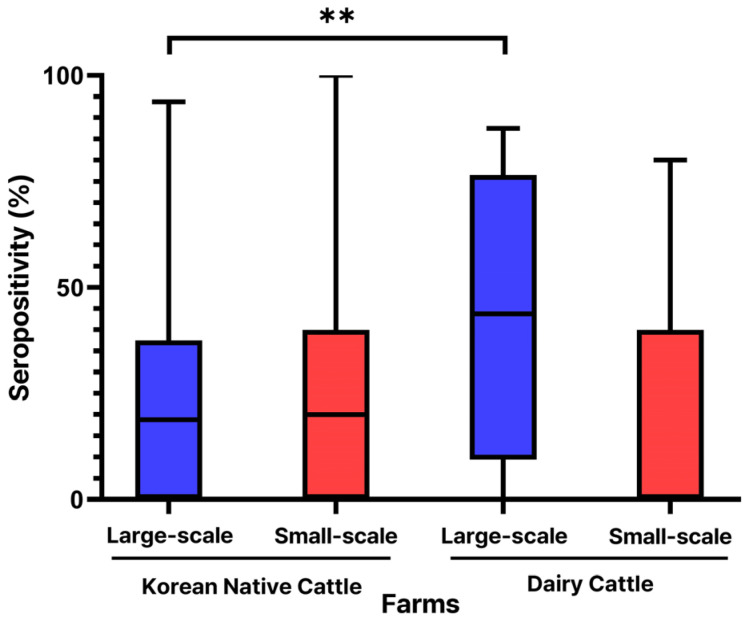
Farm scale and breed effects on seropositivity after LSD vaccination in Korean cattle. ** Statistically significant differences (*p* < 0.01).

**Table 1 animals-14-03236-t001:** Seroprevalence of lumpy skin disease virus in cattle in Korea.

Region (Provinces)	Number of Positive Cattle			Number of Positive Cattle Farms		
	Korean Native Cattle (%)	Dairy Cattle (%)	Total	Korean Native Cattle (%)	Dairy Cattle (%)	Total
Incheon	40/79 (50.63)	3/5 (60.00)	43/84 (51.19)	6/7 (85.71)	1/1 (100)	7/8 (87.50)
Jeonbuk	312/1149 (27.15)	108/216 (50.00)	420/1365 (30.77)	94/111 (84.69)	19/19 (100)	113/130 (86.92)
Jeonnam	644/2202 (29.25)	47/154 (30.52)	691/2356 (29.33)	106/214 (49.53)	6/11 (54.55)	112/225 (49.78)
Jeju	32/89 (35.96)	10/16 (62.50)	42/105 (40.00)	8/9 (88.89)	1/1 (100)	9/10 (90.00)
Total	1028/3519 (29.21)	168/391 *** (42.97)	1196/3910 (30.59)	214/341 (62.76)	27/32 * (84.38)	241/373 (64.61)

* Statistically significant differences (*p* < 0.05); *** Statistically significant differences (*p* < 0.001).

**Table 2 animals-14-03236-t002:** Posterior odds ratios for seroconversion after lumpy skin disease vaccination.

Variable	Odds Ratio		
	Mean	95% Credible Interval Lower	95% Credible Interval Upper
Large-scale farm	0.97	0.77	1.22
Dairy cattle	1.82	1.27	2.60

## Data Availability

The original contributions presented in the study are included in the article; further inquiries can be directed to the corresponding author.

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
