# Peer review of "Assessing Post-Vaccination Seroprevalence and Enhancing Strategies for Lumpy Skin Disease Vaccination in Korean Cattle"

_animals, 2024, doi:10.3390/ani14223236_

Round 1
Reviewer 1 Report
Comments and Suggestions for Authors
The manuscript titled “Assessing Post-Vaccination Seroprevalence and EnhancingStrategies for Lumpy Skin Disease Vaccination in Korean Cattle” describes LSD vaccine vaccination strategies and biosecurity measures.
Minor concerns
1. Three different live attenuated LSD vaccines, is there a gene deletion, which gene does the ELISA test for, Whether genetic deletions affect elias test results
2. Overall, 30.59% of the cattle tested (1,196 out 43 of 3,910) exhibited positive antibody responses. The protection rate of immune treasure is too low, it is recommended to strengthen the immunization
3. In addition to vaccines, enhancing biosafety is also an important means of prevention, and the authors should indicate which biosafety measures are available.
4. Dairy cattle had higher seropositivity rates (42.97%) than Korean native cattle (29.21%). Is it due to vaccine management, biosecurity, or the presence of certain antagonism in native cattle
Author Response
"Please see the attachment."

Reviewer 2 Report
Comments and Suggestions for Authors
The authors have assessed the health management in cattle farms after and for vaccination against lumpy skin disease.
The manuscript is interesting and provides suggestions for actions in a clear and significant clinical problem with global interest.
I have listed some points below that require attention to improve the manuscript before acceptance.
Introduction.
Please elaborate on the immunological response of cattle after vaccination, as this is very relevant to your study.
Materials and methods.
It is very concerning the lack of control animals. Whilst this is understandable in a project such as this one, it still requires justification and moreover, it requires specific actions to cover and compensate for the lack of controls.
Hence, 1) please justify adequately the lack of controls and 2) please describe the compensatory measures taken in the project and the analytical steps set up to consider this significant deficit.
Results.
Please colourise the graphs to improve the presentation of the final manuscript.
Discussion.
Please add a subsection with examples of health management and results from other countries of the world.
References.
The number of references is extremely small. For such a project, I expect at least 30 references in the manuscript.
Conclusions.
Please do not add new ideas in the concluding section. Please transfer these in the Discussion.
Overall.
Revision and re-evaluation.
Author Response
"Please see the attachment."

Round 2
Reviewer 2 Report
Comments and Suggestions for Authors
All the issues raised were addressed. No further comments.